# Ground Positioning Method of Spaceborne SAR High-Resolution Sliding-Spot Mode Based on Antenna Pointing Vector

**Yingying Li [1,\*], Hao Wu [1], Dadi Meng [2], Gemengyue Gao [1], Cuiping Lian [1] and Xueying Wang [1]**

[1]  Beijing Institute of Remote Sensing Information, Beijing 100192, China
[2]  Aerospace Information Research Institute, Chinese Academy of Sciences, Beijing 100190, China
[\*]  Correspondence: liyy202208@alumni.nudt.edu.cn; Tel.: +86-150-0123-1477

**Abstract:** As a new high-resolution spaceborne SAR observation mode, sliding-spot imaging has the characteristics of a large squint, long aperture time, and azimuth aliasing, and because of the dechirp operation in the imaging algorithm of this mode, it is difficult to construct a direct range–Doppler equation for its geometric processing. In this paper a conformation model based on an antenna pointing vector is presented, which fully considers the influence of the dechirp operation on the range image, starts from the relative position of the dechirped range image points and the satellite, and establishes a strict conversion model between the image coordinates and geographic coordinates using the accurate satellite–ground geometric conditions. Then the forward and reverse formulas for geometric processing of the sliding-spot mode are given based on this model. Finally, geometric calibration and positioning experiments under different conditions and field spaceborne SAR data are executed. Results show that after geometric errors caused by the SAR payload have been calibrated and other factors such as atmospheric delay, platform position, and elevation error have been compensated, the uncontrolled geometric positioning accuracy can reach within 1 m–2 m, which fully proves the effectiveness of this method in the geometric positioning of high-resolution sliding-spot images.

**Keywords:** SAR; sliding-spot mode; squint; dechirp; range–Doppler; geometric calibration; geometric positioning accuracy

## 1. Introduction

The geometric positioning accuracy of spaceborne synthetic aperture radars (SAR) is one of the key factors that determine whether the benefits of SAR image applications can be fully utilized. How to ensure the high geometric positioning accuracy of SAR is an important issue that needs to be solved urgently. Studying the geometric calibration of on-orbit SAR satellites, and applying the calibration parameter compensation to the image geometric positioning processing, is of great significance for the improvement of SAR geometric positioning accuracy.

The ERS-1 launched by ESA in 1991 is the world's first SAR satellite to achieve high-precision geometric calibration [1], which used the calibration field to calibrate the key system parameters of the uncontrolled geometric positioning, and the image plane positioning accuracy finally reached 13.18 m. From then on, some countries have actively planned and developed various technologically advanced spaceborne SAR. In particular, around 2007, Japan, Italy, Germany, and Canada successfully launched a number of SAR satellites, providing a wealth of high-quality images for relevant research units. The acquisition of high-quality images is inseparable from the support of geometric calibration and processing. The Japan Aerospace Exploration Agency carried out the geometric calibration for PALSAR on the ALOS-1 satellite, and its positioning accuracy of the strip mode reached 9.7 m under uncontrolled conditions [2]. Italy carried out the geometric calibration on the

COSMO-SkyMed images based on five calibration fields, and the uncontrolled positioning accuracy of the strip images was up to 3 m, and for the spotlight images this was up to 1 m [3]. Germany used 30 point targets placed in the 120 km × 40 km area to geometrically calibrate the TerraSAR-X satellites, and reached the positioning accuracy of 0.5 m in azimuth and 0.3 m in range [4–6]. The plane positioning accuracy of the Canada Radarsat-2 satellite is about 10 m [7,8]. After that, the geometric calibration of TanDEM-X launched by Germany, and the Sentinel-1A and Sentinel-1B launched by ESA were all undertaken by the TerraSAR-X satellite calibration team, and they continued to use the related geometric technology of TerraSAR-X and achieved good results [9–11]. In 2011, China carried out a geometric calibration study on the CRS1 SAR satellite and the positioning accuracy was better than two pixels after calibration [12]. Since 2017, many Chinese scholars have carried out geometric calibration on GF-3, and the final positioning accuracy is better than 3 m [13–17].

However, due to the staged development of SAR payloads and satellite platforms, the previous geometric calibration and positioning research is mainly applicable to traditional side-looking or small azimuth squint imaging modes. In recent years, due to its superior detection flexibility, revisit capability, and high resolution, the high-resolution, large squint sliding-spot imaging mode has shown great application potential in military and civilian applications, and has become one of the hot spots in platform development and imaging algorithm research. However, current researches are mainly focused on high-precision squint imaging algorithms [18,19], and there are very few researches related to high-precision geometric processing under large squint sliding-spot conditions. As well as the long synthetic aperture time of the high-resolution sliding-spot mode itself, the large squint imaging condition further aggravates the complexity and spatial variability of the target range history, and increases the design difficulty of the imaging algorithm. The imaging algorithm under this mode usually performs the dechirp operation on the range-compressed echo signals in the azimuth time and range frequency domain first [20,21], that is, by multiplying the dechirp factor that changes with azimuth time, so as to straighten the curved range history of all signals caused by squint and long synthetic aperture time, reduce the total doppler bandwidth and the difficulty of the imaging algorithm design. Then the traditional imaging algorithm can be performed on the dechirped output echo signal to generate a focused image [22]. However, the image is generated on the signal whose real range history has been modified after the dechirp. In this virtual range history space, the focusing and positioning of the image point are uniform and self-consistent, but the real geometric position relationship between the image point and the satellite has been destroyed by the dechirp, so the range–Doppler (RD) model cannot be directly adopted in the geometric processing of this image. How to establish a geometric processing scheme suitable for the dechirp imaging algorithm is an urgent problem to be solved.

This paper analyzes the influence of the dechirp operation on the high-resolution squint SAR sliding-spot imaging and presents a geometric constellation model based on the antenna pointing vector. The method starts from the new relative positional relationship between the SAR satellite and the image point after dechirp, and establishes a strict conversation model between image coordinates and geometric coordinates using the accurate satellite–target geometric conditions. Then the forward and reverse geometric processing flows for the calibration and positioning of the high-resolution sliding-spot mode are given. Finally, using the actual spaceborne SAR data on the calibration fields, experiments are carried out to verify the image geometric accuracy of this model. After the calibration and compensation of all the error factors, including SAR transceiver channel delay, atmospheric propagation delay, platform position error, and target elevation error, the geometric positioning accuracy can be improved to the decimeter level, which fully proves the effectiveness of our method.

## 2. Dechirp Influence on Squint Sliding-Spot Imaging Mode

We simulated the squint sliding-spot echo data of 18 point targets in the target ground area, and the SAR imaging parameters used in the simulation are shown in Table 1. Figure 1 indicates the satellite-to-ground position relationship and the distribution of targets. The satellite descends from north to south and flies in the left side looking, and performs sliding-spot imaging on the target area by turning backward with the azimuth angle from small to large.

**Table 1.** Spaceborne SAR squint sliding-spot imaging simulation parameters.

| | |
|---|---|
| Satellite orbit altitude | 600 km |
| Squint scanning angle range | 4.93–13.13 degrees |
| Center side-looking angle | 27 degrees |
| Range bandwidth | 800 MHz |
| Range sampling rate | 1000 MHz |
| Satellite platform velocity | 7.62 km/s |
| Pulse repetition frequency (PRF) | 5182 Hz |
| Imaging time | 15.26 s |
| Range sampling point number in the imaging area | 25,600 |
| Azimuth sampling point number in the imaging area | 46,272 |

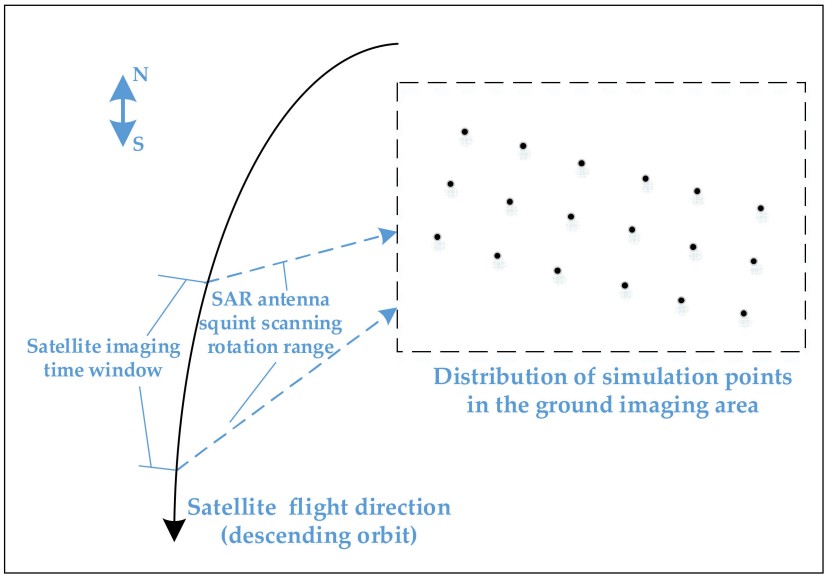

**Figure 1.** The satellite-to-ground position relationship in the simulation and the distribution of simulation points.

After the echo data are range-compressed, the SAR signal of any target can be expressed as

$$\widetilde{ss}(\tau, \eta) = p_r \left[ \tau - \frac{2r(\eta)}{c} \right] \cdot w_a(\eta) \cdot exp \left\{ -j4\pi f_0 \frac{r(\eta)}{SOL} \right\} \tag{1}$$

Here $f_0$ is the radar center frequency, $c$ is the wave speed, $\eta$ is the azimuth time, $\tau$ is the slant range delay time, $p_r()$ is the range-compressed pulse, $w_a()$ is the azimuth antenna modulation, and $r(\eta)$ is the slant range history between SAR and the target. The signal represented by this formula is shown in Figure 2. We can see that the large range of antenna rotation angle and the squint angle of view lead to serious range walking in the echo slant

range history of each target. Perform range FFT (Fast Fourier Transform) on Equation (1) to obtain

$$\widetilde{SS}(f_\tau, \eta) = W_r(f_\tau) \cdot w_a(\eta) \cdot exp\left\{ -j\frac{4\pi(f_0 + f_\tau)}{c} r(\eta) \right\} \tag{2}$$

where $W_r(f_\tau)$ is the pulse range spectral envelope.

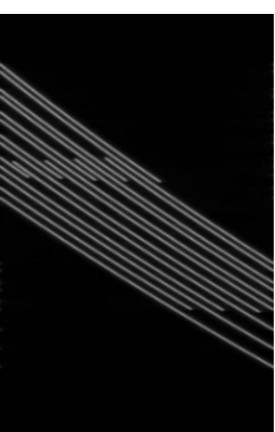

**Figure 2.** Data after range compressed.

It can be seen that in the high-resolution squint sliding-spot SAR mode, the total azimuth bandwidth may exceed several PRFs, resulting in a severe azimuth aliasing phenomenon. If no effective measures are taken, the SAR image quality will be seriously affected: on the one hand, the target focusing quality will seriously degrade; on the other hand, a large number of blurred images will be caused. Thus, compared with the traditional side-looking strip or low-resolution sliding-spot modes, the imaging algorithm for the squint sliding-spot mode has great changes. Current algorithms [20,21] commonly adopt the azimuth dechirp processing in the range frequency and azimuth time domain to correct the curved range history, move the doppler center of all signals to zero, and reduce the doppler bandwidth to a PRF, which can solve the problem of insufficient PRF. That is, perform the phase correction by multiplying Equation (2) by a chirp factor $\widetilde{Q}(f_\tau, \eta)$ as follows

$$\widetilde{Q}(f_\tau, \eta) = exp\left\{ -j\frac{4\pi(f_0 + f_\tau)}{c} D(\eta) \right\} \tag{3}$$

$D(\eta)$ in this factor is a function of $\eta$ as an independent variable, and its form is not completely fixed. It is calculated from real-time parameters such as satellite position, attitude, and antenna pointing. For example, $D(\eta)$ is defined as follows in [20]

$$\widetilde{Q}(f_\tau, \eta) = exp\left\{ -j\frac{4\pi(f_0 + f_\tau)}{c} D(\eta) \right\} \tag{4}$$

where $r_d(\eta)$ is the range history between the antenna and the rotation center, and $\overline{r_d(\eta)}$ is the mean value of $r_d(\eta)$.

The multiplication of Equations (2) and (3) is equivalent to superimposing the original slant range with $D(\eta)$, and the purpose is to eliminate the curvature of the slant range history $r(\eta)$. We can obtain

$$SS(f_\tau, \eta) = \widetilde{SS}(f_\tau, \eta) \cdot \widetilde{Q}(f_\tau, \eta) = W_r(f_\tau) \cdot w_a(\eta) \cdot exp\left\{ -j\frac{4\pi(f_0 + f_\tau)}{c} (r(\eta) + D(\eta)) \right\} \tag{5}$$

The imaging range history of each target point is straightened after dechirp by Equation (5) in the range frequency and azimuth time domain. After this step, different slant range offsets are produced on different azimuth lines, which causes the data frame to be bended in range direction and further results in a bended focused slant range image along

the azimuth direction. Then it should be neatly cropped into a square piece of echo data for the execution of subsequent imaging algorithms, and then Figure 3 is obtained. Compared with Figure 2, severe range walk and range curvature are eliminated on the new output dechirped data, and the range history of each target is symmetrical in azimuth direction and walks across a much fewer number of range gates. In addition, the doppler bandwidth of (5) is reduced less than one PRF, and the doppler center is moved to zero (with slight and acceptable offset caused by the curved orbit). On this basis, the traditional focusing imaging algorithm is performed to output the slant range image (Figure 4). After the geometric positioning process on the slant range image, the geographic projection image is obtained as Figure 5, which is the problem to be solved in this paper.

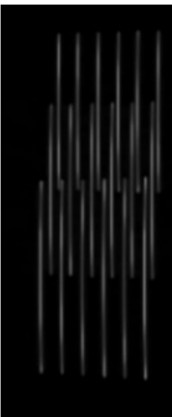

**Figure 3.** Data after dechirp and range cropping.

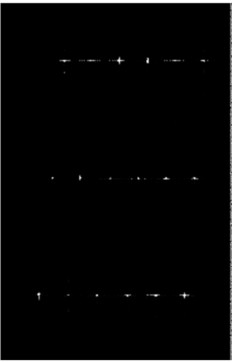

**Figure 4.** Image after focused imaging.

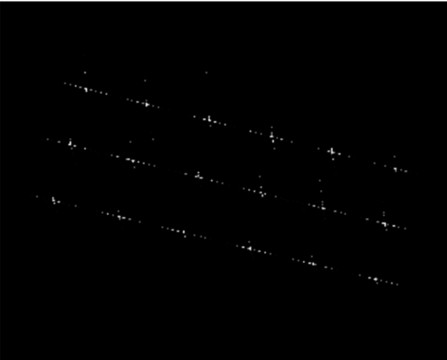

**Figure 5.** Image after geometric positioning.

### 3. Geometric Processing Principle of Spaceborne SAR High-Resolution Sliding-Spot Mode Based on Antenna Pointing Vector

*3.1. Extension of the Range–Doppler Model after Dechirp*

The classical geometric positioning model—RD equation—is as follows [23], which associates the target position vector with the position and velocity vector of the SAR satellite at a certain azimuth moment. At this moment, the observation geometry between SAR and the target is determined by the distance and the doppler frequency of the target from SAR. This model closely fits the geometric property of SAR imaging, and can perform absolute positioning toward any image pixel without control points, so it is widely used in the system-level geometric processing of spaceborne SAR.

$$
\begin{cases}
\text{Range equation: } r = \left| \vec{S} - \vec{T} \right| \\
\text{Doppler equation: } f_d = -\dfrac{2}{\lambda R} \left( \vec{S} - \vec{T} \right) \cdot \left( \vec{V_S} - \vec{V_T} \right) \\
\text{Earth model equation: } \dfrac{T_x^2 + T_y^2}{(R_e + h)^2} + \dfrac{T_z^2}{R_p^2} = 1
\end{cases}
\tag{6}
$$

$\vec{S}$ is the SAR antenna center position vector and its velocity vector is $\vec{V_S}$. $\vec{T} = (T_x, T_y, T_z)$ is the ground position vector of the target point, and its velocity vector is $\vec{V_T}$. $r$ represents the range between the target point and SAR, and $\lambda$ is the echo center wavelength. $f_d$ is the doppler center frequency at this azimuth moment, which is output by the imaging processing. $R_e$ and $R_p$ are the Earth ellipsoid parameters.

The RD equations can be directly applied in the geometric processing of the side-looking strip or low squint mode. However, high-resolution or squint SAR modes must be achieved through a large range of the azimuth scanning angle, which means the azimuth aliasing phenomenon under this condition may be very complicated and the dechirp operation before the focusing imaging process is necessary. As shown in Section 2, the key step of dechirp is to multiply the range-compressed data within the azimuth time and range frequency domain by a slant range bias factor $\widetilde{Q}(f_\tau, \eta)$, which varies non-linearly along the azimuth direction so that the curved range history of each target becomes symmetrical and straight.

As shown in Figure 6, the SAR satellite antenna rotates through an angular range around the virtual rotation point and continuously observes a certain target during sliding-spot imaging. The curved range history of the target echo data after range compression is shown in Figure 7a. After dechirp, the range history is straightened as in Figure 7b, and then the data are neatly cropped along the azimuth direction as in Figure 7c. Due to the time domain offset introduced by dechirp, the near range of each azimuth line on the new one-dimensional range image (Figure 7c) varies along the azimuth direction. Then the imaging algorithms can be executed on this dechirped new data to generate an image.

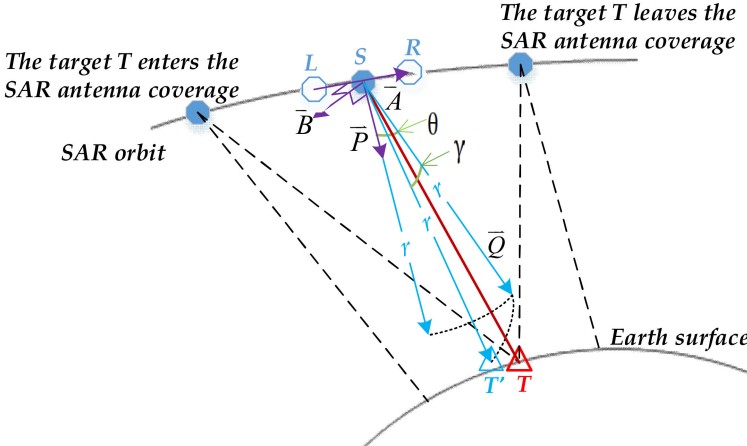

**Figure 6.** Squint sliding-spot imaging of target by the SAR satellite and the geometric forward process based on antenna pointing vector.

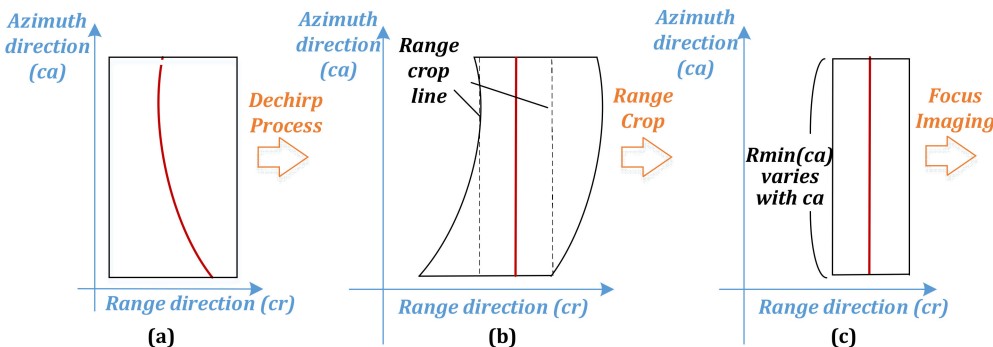

**Figure 7.** (**a**) Curved range history of the target; (**b**) after dechirp; (**c**) after range crop along azimuth.

Since the original geometric relationship between the image point and the satellite has been destroyed during the dechirp process, the main purpose of this paper is to re-establish the geometric relationship between the image and the satellite so as to propose a geometric processing scheme suitable for the dechirp imaging algorithm. As validated in [20], after dechirp processing, the range history of each target is shifted to be symmetrical in the azimuth direction, and the doppler center is moved to zero. Thus, there are two characteristics of this image output in the virtual range history space after dechirp: one is that the range history of any target is symmetrical and straight relative to the satellite position at the imaging azimuth moment of the target; the other is that on the entire satellite orbit, the distance between the target and the satellite at this moment is the shortest. These two items are the basis for the geometric design in the next section. It should be noted that each dechirped aperture of the dechirped data is eventually focused to the location with the closest range position and zero doppler frequency, the geometric positioning is also performed on this new image, so even if there is a non-ideal attitude error that causes the doppler frequency after dechirp to be non-zero, it will only cause the decrease in the image radiation focusing quality, and no position error will arise because of this.

The image coordinates of the target on the slant range image are known, the process of solving its geographic coordinates is called geometric forward calculation, and this process is mainly used for the geometric positioning. The latitude and longitude coordinates of the target are known, the process of solving its image coordinates is called geometric inverse calculation, and this process is mainly used for geometric calibration. Based on the upper two characteristics of the new dechirped range image, combined with the range equation in (6), respectively, this paper constructs an accurate satellite–ground position relationship suitable for the squint sliding-spot imaging and proposes a geometric process including forward and inverse calculation.

*3.2. The Geometric Forward Calculation Process*

As in Figure 6, the SAR satellite images the ground area using the squint sliding-spot mode, and after the squint imaging process described in Section 2, a slant range image is output, which does not contain geographic information. Based on the image coordinate of any target point, we obtain the slant range and the satellite position at the azimuth time of the target. Then we can calculate the azimuth squint angle of the antenna toward the target, and combine with the slant range and the target elevation to calculate the real central pointing vector at the target. Finally, the intersection point of this pointing vector and the earth surface is obtained as the geographic coordinates of the target.

(a) The image coordinate $(cr, ca)$ of the target point $T$ is known, and $cr$ is the range row number. Because the dechirped data are eventually focused to the location with the zero doppler frequency, the zero doppler moment of the target image point after dechirp is taken as the imaging moment of the point. Then a one-to-one correspondence between the azimuth line number $ca$ of the image and the azimuth time $\eta$ can be established. Calculate the slant range of $(cr, ca)$ as

$$r = r_{min}(ca) + cr * c / (2 * f_S) \tag{7}$$

Here $r_{min}(ca)$ is the real near range of line $ca$ in the slant range image obtained after dechirp and cropping, which is determined by the original near range before dechirp and the offset $D(\eta)$ in Equation (3) jointly. Because the slant range offset of each image line is different during the dechirp process, $r_{min}(ca)$ changes with $ca$. Suppose the surface elevation of the target $T$ is $H$, $f_S$ is the range sampling frequency of SAR, $\gamma$ is the range side-looking angle, which refers to the angle between the antenna pointing and the satellite orbital plane. Now we begin to solve the latitude and longitude coordinates $(T_{lat}, T_{lon})$ of $T$.

(b) Suppose SAR locates at point $S$ at time $ca$, and its position coordinates in the WGS 84 coordinate system are $(S_x, S_y, S_z)$. Extend the moment forward and backward at equal time intervals to obtain the adjacent moments $ca - c1$ and $ca + c1$; $c1$ is the change row amount corresponding to the time interval, generally taking 3 to 5. Suppose SAR locates at point $L$ and $R$ at $ca - c1$ and $ca + c1$; its position coordinates are $(R_x, R_y, R_z)$ and $(L_x, L_y, L_z)$, respectively. The near slant range of the corresponding lines are, respectively, $r_{min}(ca - c1)$ and $r_{min}(ca + c1)$. Points $L$, $R$, and $T$ form a triangle $\Delta LTR$ whose vertex is $T$ and base is $LR$. Since the $c1$ value is very small, $|LR| \ll |ST|$, so $S$ can be approximately taken as the midpoint of $LR$, then the half distance of the base is $c = |LR|/2$, and the length of the triangle midline on the base is $|ST| = r$. As described above, the range history of any target on the slant range image after dechirp is symmetrical and straight, so the difference $d$ between the two triangle waist lengths is the difference between two slant offsets of the line $ca - c1$ and $ca + c1$, and the calculation equation is

$$d = |LT| - |RT| = r_{min}(ca - c1) - r_{min}(ca + c1) \tag{8}$$

(c) According to the triangle midline theorem, knowing the half distance $c$ of the base, the difference $d$ between the lengths of the two waists, and the length $r$ of the base midline, we can calculate the angle $\theta$ between the midline and the vertical line of the base as $\theta = arccos\left[\frac{d \cdot \sqrt{r^2 + c^2 - d^2/4}}{2 \cdot r \cdot c}\right] - \frac{\pi}{4}$ (Figure 8), which is exactly the squint angle of SAR at point $S$. The squint angle refers to the angle between the antenna pointing and the plane perpendicular to the satellite's motion direction.

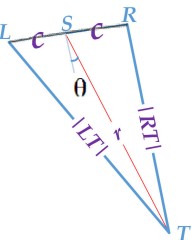

**Figure 8.** Solving the squint angle of antenna pointing using the triangle midline theorem.

(d) The unit vector of the satellite motion direction at point $S$ is calculated as $\vec{A}$ $(A_x, A_y, A_z) = Normalize(R_x - L_x, R_y - L_y, R_z - L_z)$; here $Normalize(\cdot)$ represents the vector normalization operation.

(e) On the plane passing through point $S$ and perpendicular to vector $\vec{A}$, we construct a unit vector $(P_x, P_y, P_z)$, which is defined as follows

$$
\begin{cases}
If\ max(|A_x|, |A_y|, |A_z|) = |A_x|,\ then \begin{cases} P_y = -S_y \\ P_z = -S_z \\ P_x = -(A_yP_y + A_zP_z)/A_x \end{cases} \\
If\ max(|A_x|, |A_y|, |A_z|) = |A_y|, then \begin{cases} P_x = -S_x \\ P_z = -S_z \\ P_x = -(A_xP_x + A_zP_z)/A_y \end{cases} \quad => Get\ normalized\ \vec{P} = Normalize(P_x, P_y, P_z). \quad (9) \\
If\ max(|A_x|, |A_y|, |A_z|) = |A_z|, then \begin{cases} P_x = -S_x \\ P_y = -S_y \\ P_z = -(A_xP_x + A_yP_y)/A_z \end{cases}
\end{cases}
$$

The vector $\vec{P}$ is defined as pointing to the center of the earth as much as possible. In order to avoid the situation that $\vec{P}$ is abnormal because an element of $\vec{A}$ is close to 0, the maximum value among the three elements of $\vec{A}$ is used as the conditional factor for constructing $\vec{P}$, so it needs to be defined in three cases.

(f) The plane $ASP$ passes through point $S$, and vector $\vec{A}$ and $\vec{P}$. The unit vector $\vec{B}$ is set to be perpendicular to the plane $ASP$. Then $\vec{A}$, $\vec{B}$, $\vec{P}$ form three-dimensional unit vectors perpendicular to each other in the space under the WGS84 coordinate system.

(g) Let the starting point of the iteration vector $\vec{Q}$ be point $S$, its initial pointing be $\vec{P}$, and its length $\left|\vec{Q}\right|$ be the slant range $r$ from SAR to the target. First rotate $\vec{Q}$ around $\vec{B}$ by $\theta$ degree, then around $\vec{A}$ by $\gamma$, and obtain point $T\prime = \left(T'_x, T'_y, T'_z\right)$ as the end of $\vec{Q}$. Convert the coordinates into latitude, longitude, and elevation coordinates to obtain $\left(T'_{\text{lat}}, T'_{lon}, H'\right)$. According to the surface elevation $H$ of the target $T$, obtain an updated point $T$ as $\left(T'_{\text{lat}}, T'_{lon}, H\right)$. Calculate the distance $|ST|$ between point $S$ and $T$, and find the projection length $m$ of vector $\vec{ST}$ on unit vector $\vec{B}$ and the projection length $n$ on unit vector $\vec{P}$. Then the side-looking angle $\gamma$ can be updated as

$$\gamma = atan(m/n) \tag{10}$$

Here the rotation angle $\theta$ is fixed in each loop, in order to ensure that the search is carried out in the zero Doppler plane after dechirp, and the target point that satisfies the range value $r$ is searched out by rotating the side-looking angle $\gamma$.

(h) If $||ST| - r| > 10^{-6}$, return to (g) and continue the loop. Otherwise, it means that $|ST|$ is consistent with the real slant range $r$ of target $T$, and the updated coordinates $\left(T'_{\text{lat}}, T'_{lon}, H\right)$ can be taken as the positioning result of target $T$, and then the loop ends. The flowchart of the geometric forward process is shown in Figure 9.

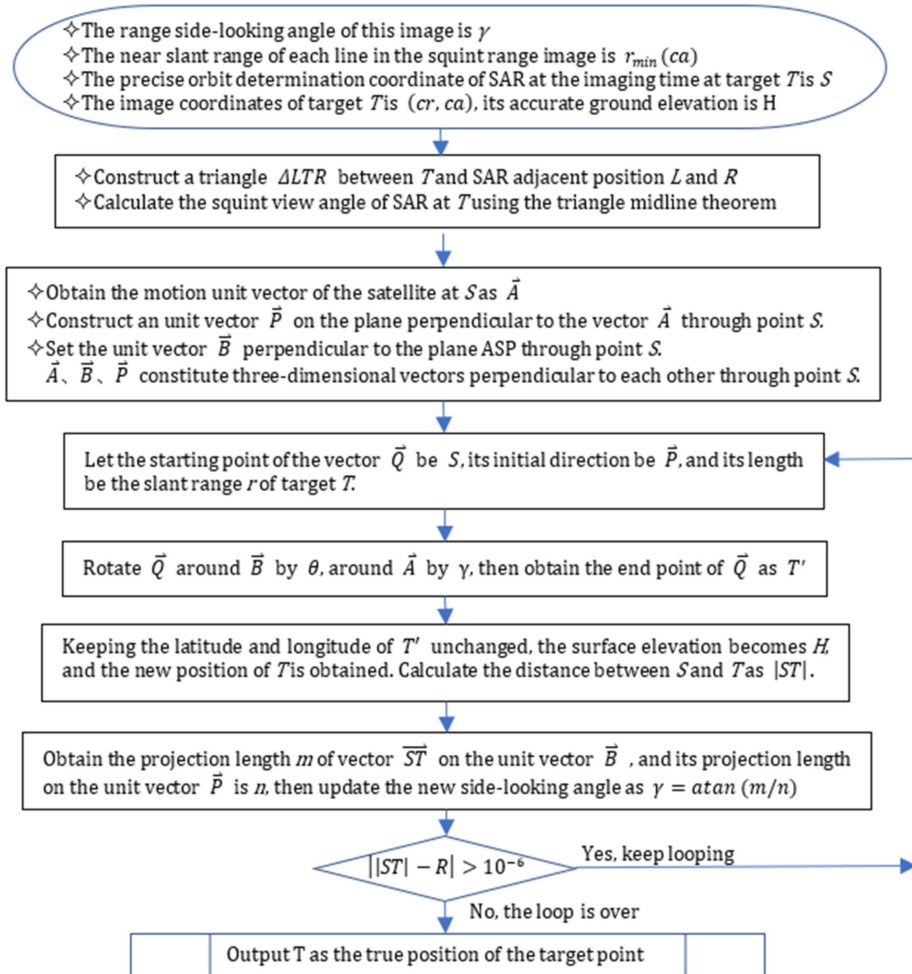

**Figure 9.** Forward geometric positioning flow for squint sliding-spot imaging mode.

*3.3. The Geometric Inverse Calculation Process*

As mentioned above, the doppler centers of all targets are pulled to zero frequency after dechirp. That is, for a certain target on the new slant range image (Figure 3), in the entire satellite flight trajectory, the moment when the satellite is closest to any target point is the azimuth moment of the target imaging. Based on the geographical coordinates of the target and the entire SAR trajectory, we can obtain a certain SAR position and corresponding imaging time when the slant range between SAR and the target is closest (removing the slant range offset) through iteration. Then the image pixel coordinates of the target can be calculated according to the imaging time and the closest slant range.

Knowing the latitude and longitude coordinates $(T_{lat}, T_{lon})$ and elevation $H$ of the target $T$, the process of obtaining its image coordinates $(cr, ca)$ is as follows:

a) Convert $(T_{lat}, T_{lon}, H)$ to 3D coordinates $(T_x, T_y, T_z)$ under WGS84.

b) Set the initial value $ca_{begin} = 0$, $ca_{end} = m\_nHeight$. m_nHeight is the total number of lines in the azimuth direction of the slant range image.

c) Let $ca = (ca_{begin} + ca_{end})/2$. Suppose the satellite locates at point $S_{ca}$ and $S_{ca-1}$ at time $ca$ and $ca - 1$, respectively. Calculate the distances between the two points and $T$, respectively, and remove the near slant range to find the difference between the two $N = \{|S_{ca}T| - r_{min}(ca)\} - \{|S_{ca-1}T| - r_{min}(ca-1)\}$.

d) If $N > 0$, it indicates that the target azimuth coordinate is between $ca_{begin}$ to $ca$, let $ca_{end} = ca$. Otherwise, it indicates that the target azimuth coordinate is between $ca$ to $ca_{end}$, let $ca_{begin} = ca$.

e) If $ca_{end} - ca_{begin} > 1$, go to (c) to execute a new loop, otherwise the loop ends, and $ca$ is the azimuth image coordinate of the target $T$, and its range image coordinate is calculated as $cr = (|S_{ca}T| - r_{min}(ca)) * 2 * f_S/c$.

## 4. Experiment of Geometry Processing Accuracy Evaluation

The geometric processing method for the spaceborne SAR high-resolution sliding-spot mode based on the antenna pointing vector proposed in this paper consists of two parts: the forward process of calculating the geographic coordinates from the image coordinates of the target and the inverse process of calculating the image coordinates from the geographic coordinates. In order to effectively evaluate the effectiveness of the above methods, this section conducts two-step geometric processing experiments based on the high-resolution sliding-spot data obtained by the same SAR satellite under different pulse-width and bandwidth combinations and within different calibration fields: In the first step, the geometric inverse calculation method is used to perform the calibration process and obtain accurate geometric calibration parameters; in the second step, on the basis of high-precision geometric calibration, the geometric forward calculation method is used to perform geometric positioning and accuracy verification. The experimental data used in these two steps are not related to each other.

The factors of spaceborne SAR geometric positioning errors always include four parts:

One is the orbital positioning error of the SAR satellite, which leads to deviations in the range and azimuth directions. At present, most SAR satellites provide precise orbit determination products based on post-event precise ephemeris calculations, and their accuracy can reach *cm* level [24,25], which is adopted in our experiment.

Second, the error of atmospheric propagation delay leads to the deviation in slant range measurement [26]. Here, according to the NCEP global atmospheric parameters updated every six hours and the global TEC data provided by CODE, the correction values for the neutral atmospheric propagation delay and ionospheric delay are calculated, respectively, from the antenna phase center to the image center point, in order to eliminate the influence of atmospheric propagation delay for this image. The atmospheric delay correction models adopted here can be obtained as [27,28].

The third is the elevation error of the check points, which leads to the deviation in the range and azimuth. In our experiment, both the high-precision elevation data measured in the field and the elevation data extracted from the different scale DEM data are used to verify the influence of elevation errors on the geometric positioning accuracy.

The fourth is the range and azimuth errors caused by the deviations in the slant time delay and azimuth time unify of the SAR payload. On the basis of stripping away the above three factors as much as possible, geometric calibration is adopted to solve these two compensation parameters.

### 4.1. Introduction of Experimental Data

The experiments were carried out on the Chinese Remote Sensing Satellite, but the method is also applicable to other high-resolution SAR satellites such as QL-1. Eighteen sliding-spot scenes of the China Xinjiang calibration field were collected as calibration images. It should be noted that both the range sampling frequency (RSF) and the pulse-width of the SAR signal will affect the range measurement accuracy [12]. The higher the RSF, the higher the ranging accuracy, and the RSF and the range bandwidth are completely correlated. The pulse-width also affects the ranging accuracy because different reference points for recording the system delay results in different system delays. Therefore, for a certain satellite, the geometric calibration and positioning process need to be executed according to different combinations of the pulse-width and bandwidth parameters of the SAR system. Here the 18 images are divided into 6 calibration groups B1–B6 according to their pulse-width and bandwidth parameter combinations. Their imaging information is shown in Table 2. We use several corner reflectors as high-precision ground control points for the calibration process. Their geographic coordinates are obtained by GPS measurement

whose accuracy is better than 0.1 m. The image coordinates of these reflectors are obtained by manual selection whose accuracy is better than 1 pixel.

**Table 2.** Incalib image information.

| Pulse-Width and Bandwidth Combination ID | Incalib Image ID | Central Side-Looking Angle | Orbit | Looking Side |
|---|---|---|---|---|
| | Image 1 | 50.27 | Ascend | Right |
| B1 | Image 2 | 47.93 | Descend | Left |
| | Image 3 | 47.43 | Descend | Left |
| B2 | Image 4 | 28.44 | Descend | Left |
| | Image 5 | 54.92 | Descend | Left |
| | Image 6 | 52.11 | Ascend | Right |
| | Image 7 | 53.21 | Descend | Left |
| B3 | Image 8 | 53.21 | Ascend | Left |
| | Image 9 | 44.96 | Ascend | Left |
| | Image 10 | 50.7 | Ascend | Right |
| | Image 11 | 58.86 | Ascend | Right |
| B4 | Image 12 | 39.37 | Ascend | Left |
| | Image 13 | 23.62 | Ascend | Right |
| | Image 14 | 18.33 | Descend | Right |
| B5 | Image 15 | 31.39 | Ascend | Right |
| | Image 16 | 29.44 | Ascend | Left |
| | Image 17 | 23.62 | Ascend | Left |
| B6 | Image 18 | 57.2 | Descend | Left |

Fourteen sliding-spot images of Chinese Inner Mongolia calibration field from the same SAR satellite were collected as verification data, and the geometric positioning was processed to verify the uncontrolled positioning accuracy after geometric calibration based on the above calibration images. In this area, several corner reflectors are also laid out as high-precision check points. The coordinate measurement methods of these points are the same as the calibration control points. Corresponding to pulse-width and bandwidth combinations of the calibration groups, the verification images are also divided into five verification groups B1–B5 whose image information is shown in Table 3.

**Table 3.** Test image information.

| Pulse-Width and Bandwidth Combination ID | Test Image ID | Central Side-Looking Angle | Orbit | Looking Side |
|---|---|---|---|---|
| | Image 1 | 48.43 | Descend | Left |
| B1 | Image 2 | 48.18 | Descend | Right |
| | Image 3 | 51.72 | Ascend | Left |
| | Image 4 | 28.94 | Ascend | Left |
| B2 | Image 5 | 28.94 | Descend | Right |
| | Image 6 | 21.32 | Ascend | Right |
| | Image 7 | 50.7 | Ascend | Left |
| B3 | Image 8 | 49.14 | Ascend | Left |
| | Image 9 | 34.99 | Ascend | Left |
| B4 | Image 10 | 34.56 | Descend | Right |
| | Image 11 | 42.16 | Ascend | Left |
| | Image 12 | 29.94 | Descend | Left |
| B5 | Image 13 | 25.27 | Ascend | Left |
| | Image 14 | 19.54 | Descend | Right |

*4.2. Geometric Calibration Results*

The calibration is performed, respectively, on the scenes in Table 2. For example, Figure 10 shows the range and azimuth deviations of all corner reflectors in Incalib Image

11 and Incalib Image 17 after the compensation of other errors including the atmosphere delay, platform position error, and elevation error. The circles in the figure represent the position of the calibration corner reflectors within the image scene, every red line represents the range deviation of each reflector, the blue line represents its azimuth deviation, and the length of the line represents the deviation amount. It can be seen that the value and direction of the range deviations are consistent among different corner reflectors. However, the azimuth deviation is very small, almost negligible for some scenes. We calculate the medium error of the deviations of all corner reflectors in each scene, then obtain the range meter and azimuth time correction values as the range and azimuth calibration value of this scene.

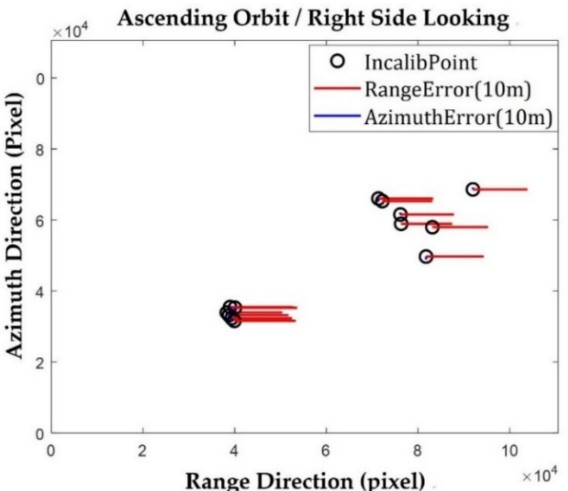 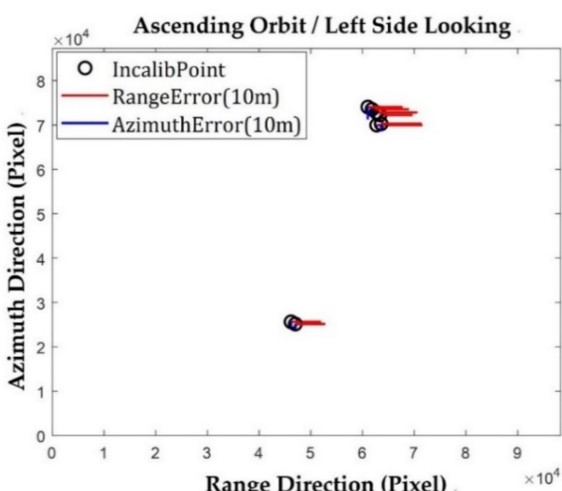

**Figure 10.** Range and azimuth calibration deviations of all corner reflectors inside a single image (Left: Incalib Image 11; Right: Incalib Image 17).

Figures 11 and 12 show the calibration values of the 18 images in Table 2. Different combination groups (B1–B6) are distinguished by different colors. Among them, the three groups B2, B4, and B6 have only one calibration image, and the three groups B1, B3, and B5 correspond to multiple images. From Figure 10 we can see that the slant range calibration values within each group B1, B3, and B5 are not stable and constant, there are random errors. However, apart from occasional outliers such as Incalib Image 3, Image 10, and Image 15, the range correction values within each group basically float within the range of 1 m–2 m. With the increase in the calibration images, the calibration results of different scenes in the same group show a stable trend. Figure 12 shows that compared with the range correction value shown in Figure 11, the impact of the azimuth time perturbation on the azimuth positioning accuracy is as small as submeter level, but it shows isotropy, which proves that they still have a certain regularity and can be used for azimuth calibration. This deviation is mainly due to the PRF timing error. Subsequent positioning verification experiment results in Section 4.3 also prove this, and the positioning error after the azimuth calibration can be reduced to a certain extent.

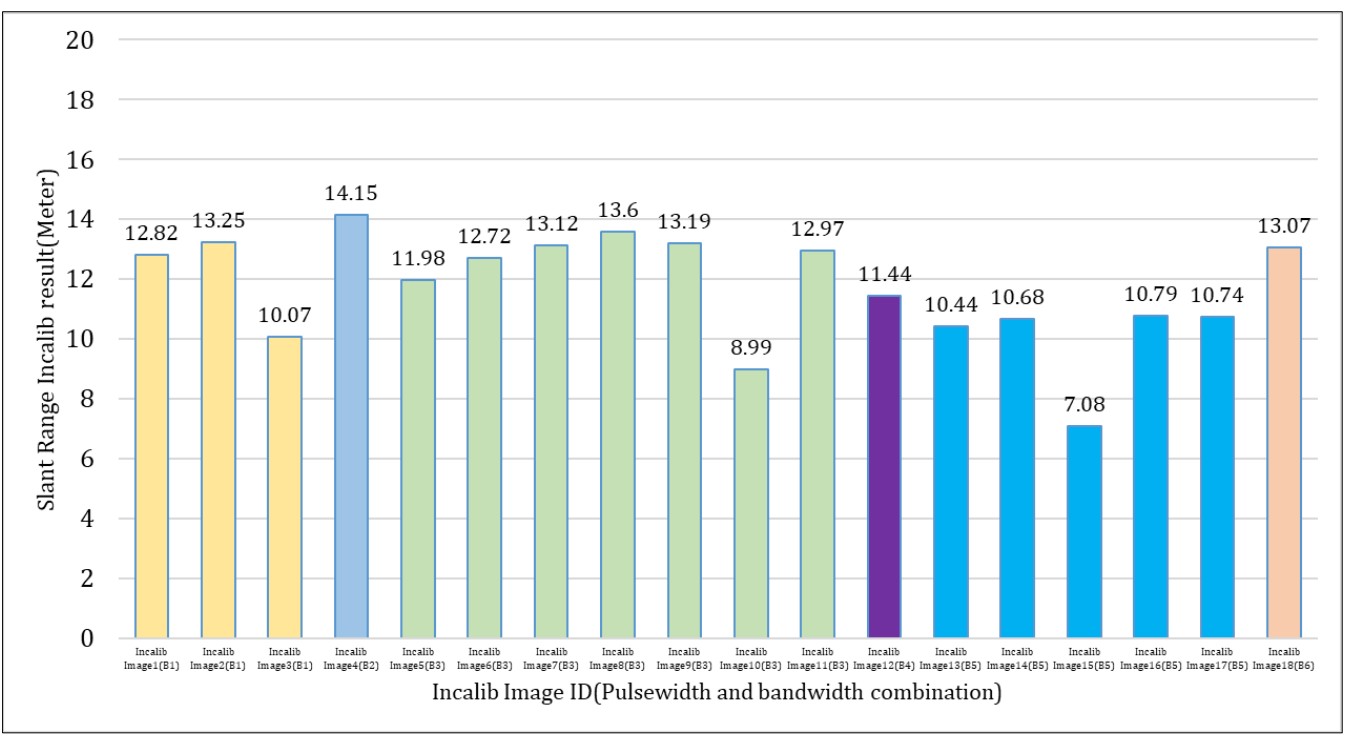

**Figure 11.** Slant range calibration values of 18 Incalib images.

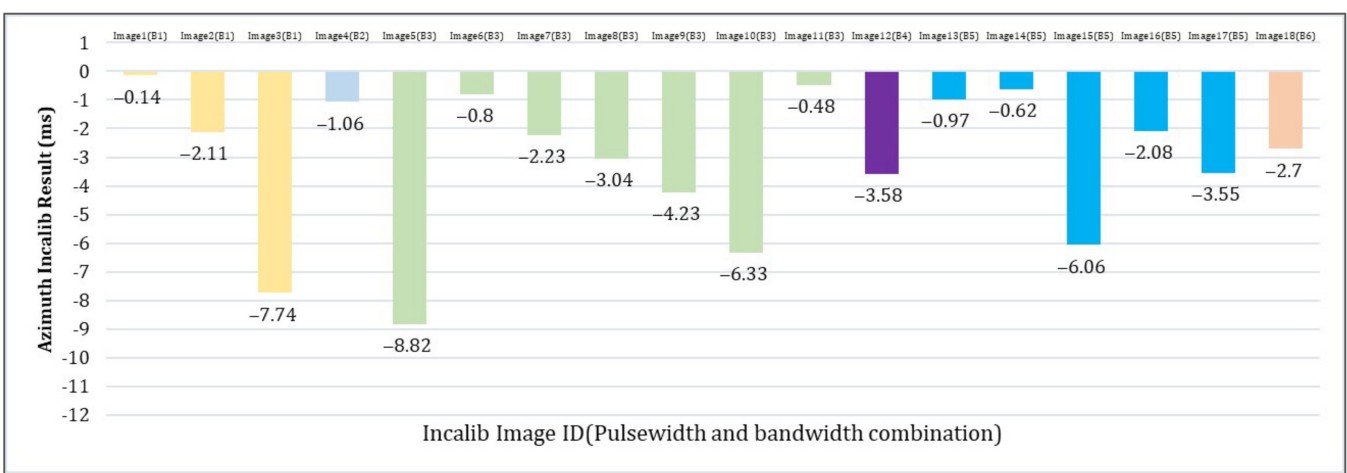

**Figure 12.** Azimuth time calibration values of 18 Incalib images.

In order to weaken the influence of random errors on the geometric calibration accuracy, we use the average of the calibration values of multiple images in the same pulse-width and bandwidth combination as the geometric calibration result for this combination. The combinations B1, B3, and B5 correspond to multiple calibration images, so Table 4 summarizes the final calibration results of these three combinations. In the table, Incalib 1 indicates the calibration result that all images included in the combination participate in the average calculation, and Incalib 2 indicates the calibration result after excluding abnormal Image 3, Image 10, and Image 15, and the rest of the images participate in the average calculation. It can be seen that the calibration results are different between different pulse-width and bandwidth combinations, and all the subsequent geometric positioning and accuracy verification experiments are carried out toward these three combinations.

**Table 4.** Calibration results under different pulse-width and bandwidth combinations.

| Pulse-Width and Bandwidth Combination | The Number of Images Involved in the Calibration | Incalib 1 (No Image Removed) | | Incalib 2 (Abnormal Images Removed) | |
|---|---|---|---|---|---|
| | | Slant Range Calibration Result (m) | Azimuth Time Calibration Result (ms) | Slant Range Calibration Result (m) | Azimuth Time Calibration Result (ms) |
| B1 | 3 | 12.046 | 3.333 | 13.035 | 1.126 |
| B3 | 7 | 12.367 | 3.708 | 12.93 | 2.160 |
| B5 | 5 | 9.946 | 2.661 | 10.662 | 1.809 |

### 4.3. Geometric Positioning Results after Calibration

Firstly, the geometric positioning results of a single scene after calibration are introduced. Take Test image 3 in Table 3 as an example, there are nine corner reflectors distributed within the scene. As shown in Figure 13, without any error correction, the original positioning average error is 18.64 m in the range direction and $-1.63$ m in the azimuth direction. After the atmospheric delay correction, the absolute positioning error in the range direction is improved to 13.43 m. Finally, after compensation using range and azimuth calibration results from Table 4, the absolute positioning error in the range direction is $0.39 \pm 0.78$ m, and $-0.63 \pm 1.31$ m in the azimuth direction.

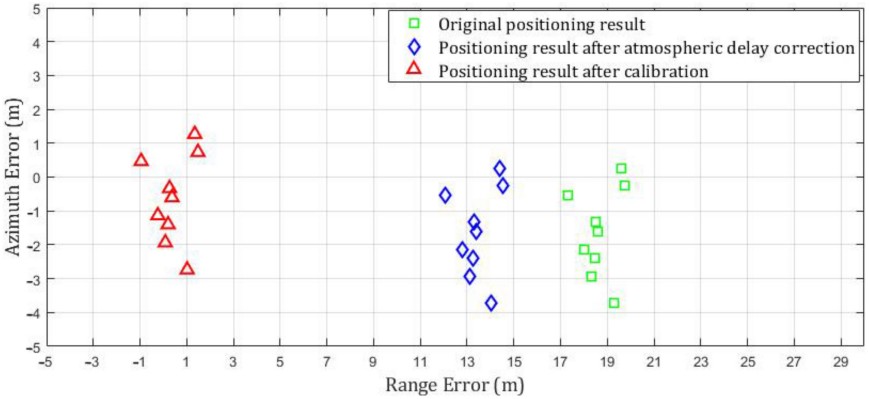

**Figure 13.** Positioning results of Test image 3 without any error correction, after atmospheric delay correction and after calibration compensation.

The above is the single image positioning result. Now the geometric positioning process is carried out for all the test images included in the combinations B1, B3, and B5 of Table 3. The range and azimuth positioning error of the original, after atmospheric delay correction, and after calibration, respectively, are shown in Figure 14. The calibrations are carried out separately using the two sets of calibration results (Incalib 1 and Incalib 2) given in Table 4. In the bottom figure of Figure 14, it should be noted that since the atmospheric delay mainly affects the range deviation, the two curves of the original azimuth error and the azimuth error after atmospheric correction are coincident. From the accuracy verification results in the figure, the following conclusions can be drawn: (1) If high geometric positioning accuracy is required, all error factors must be taken into account in the geometric processing including the echo atmospheric propagation delay, the satellite platform position error, and the SAR channel delay. After all factors are compensated, the geometric positioning errors of all the images can be controlled within 1–2 m, which fully proves the effectiveness of the geometric algorithm presented in this paper. (2) There are random and occasional anomalies in the single-image calibration values, so the final calibration results must be concluded from comprehensive analysis of multiple calibration images. (3) Calibration greatly improves the geometric positioning accuracy from tens of meters to decimeter level, and the calibrated system error has a certain difference among

different pulse-width and bandwidth combinations. (4) After eliminating various errors, there is still a positioning error of about 1m left, which is suspected to be caused by a small random error of the SAR channel delay.

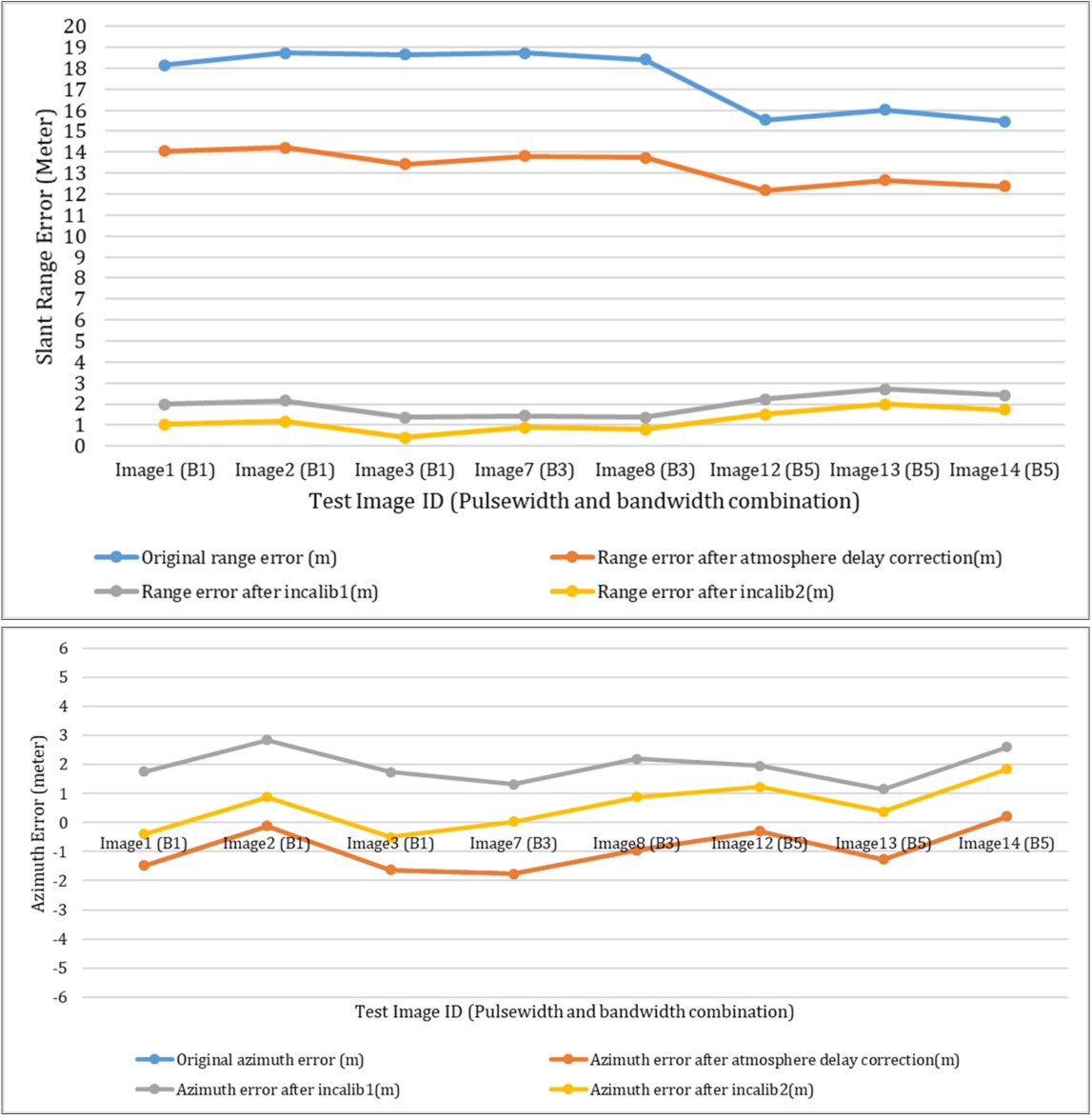

**Figure 14.** Positioning results of the test images of three groups B1, B3, and B5 (Top: range direction; Bottom: azimuth direction).

*4.4. Influence of Different Precision Elevation Data on Positioning Accuracy*

The elevations are extracted from three data sources including the field measurement, the 1km grid Global DEM, and the ASTER DEM [29]. Compared with the high-precision measurement elevation, the elevation errors of the other two DEM sources at the deployed nine corner reflectors are shown in Table 5. It can be seen that the results are consistent with the nominal accuracy of these DEM sources.

**Table 5.** Elevation errors of different elevation sources.

| Corner Reflector ID | Field Measurement (m) | ASTER DEM (m) | Global DEM (m) |
|:---:|:---:|:---:|:---:|
| 1 | 1087.467 | 1098 | 1094 |
| 2 | 1068.961 | 1087 | 1101 |
| 3 | 1091.221 | 1108 | 1122 |
| 4 | 1090.578 | 1096 | 1090 |
| 5 | 1089.080 | 1097 | 1107 |
| 6 | 1091.104 | 1104 | 1121 |
| 7 | 1092.414 | 1095 | 1121 |
| 8 | 1092.992 | 1099 | 1123 |
| 9 | 1093.143 | 1104 | 1119 |
| Medium error | | ±11.240 | ±24.984 |

We choose three images of different side-looking angles (from 19.54° to 51.72°) from Table 3 to verify their positioning accuracy under different DEM data sources. As shown in Table 6, the DEM error mainly affects the positioning accuracy in the range direction, and with the improvement in the DEM accuracy, the better the geometric positioning result will be. Therefore, obtaining high-precision DEM data and performing orthorectification on the entire SAR image is one of the keys to improving its overall positioning accuracy.

**Table 6.** Positioning accuracy of different side-looking angles under different precision DEM.

| ID | Side Looking | Elevation Source | Positioning Error | | |
|:---:|:---:|:---:|:---:|:---:|:---:|
| | | | Range | Azimuth | Plane |
| Test Image 14 | Left view 19.54 degree | Field measurement | 1.721110 | 0.218848 | 1.734968034 |
| | | ASTER DEM | 11.113978 | 0.594016 | 11.12984106 |
| | | Global DEM | 22.464659 | 1.344352 | 22.50484806 |
| Test Image 9 | Left view 34.99 degree | Measured elevation | 1.118257 | −2.620450 | 2.849080013 |
| | | ASTER DEM | 8.961166 | −2.861411 | 9.406921334 |
| | | Global DEM | 18.429528 | −3.102371 | 18.68882576 |
| Test Image 3 | Left view 51.72 degree | Measured elevation | 0.396214 | −1.631593 | 1.679011987 |
| | | ASTER DEM | 5.514775 | −1.750255 | 5.785856537 |
| | | Global DEM | 11.693389 | −1.898581 | 11.84651662 |

We re-sort the test images in Table 3 according to the side-looking angle from large to small, and the relationship between the range positioning error under the field measurement elevation input and the side-looking angle is shown in Figure 15. As the side-looking angle decreases, the positioning accuracy is lower. This is because the smaller the side view angle, the more serious the image deformation caused by the same elevation error, which is consistent with the change trend in Figure 15.

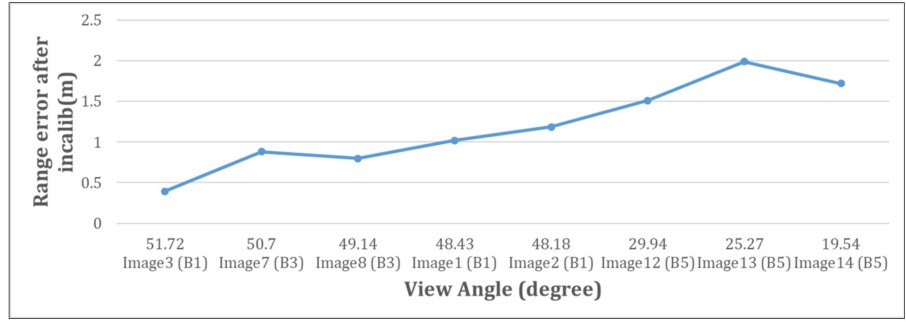

**Figure 15.** The relationship between side-looking angle and range positioning accuracy under field measurement elevation input.

## 5. Conclusions

Due to the special operation of the imaging processing algorithm for the high-resolution sliding-spot SAR mode, the traditional RD model cannot be adopted directly for its geometric processing. This paper illustrates the influence of the dechirp operation on the slant range image first and concludes the new position relationship between the output dechirped range image and SAR. Then a constellation model based on the antenna pointing vector is proposed, and the forward and the inverse calculation flows are further designed for the geometric calibration and positioning, respectively. Finally, the geometric experiments are carried out using the outfield SAR satellite data. The influence of different error factors on the geometric positioning accuracy are quantitatively given sub-item, and it is proved that after the compensation of all the errors, the image uncontrolled positioning accuracy can reach within 2 m. In addition to this, based on the experimental results, a series of suggestions are given about the reference elevation input selection, the calibration scheme design, and the error source compensation, which provide a complete and reliable basis for the design of geometric processing in the sliding-spot imaging mode. In the next research work, we will further analyze the residual error of the above geometric positioning and its influencing factors in detail so as to improve the positioning algorithm. Meanwhile, we will try to provide theoretical support for the demonstration of higher performance requirements for on-board devices on future satellites.

**Author Contributions:** Writing—original draft preparation, investigation, Y.L.; methodology, conceptualization, and project administration, H.W., D.M., and G.G.; data curation, and writing—review and editing, C.L. and X.W. All authors have read and agreed to the published version of the manuscript.

**Funding:** This research was funded by National Natural Science Foundation of China NFSC (61971047).

**Data Availability Statement:** Not applicable.

**Conflicts of Interest:** The authors declare no conflict of interest.

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
