# Peer review of "Ground Positioning Method of Spaceborne SAR High-Resolution Sliding-Spot Mode Based on Antenna Pointing Vector"

_remotesensing, doi:10.3390/rs14205233_

Round 1
Reviewer 1 Report
This paper proposes a ground positioning method for high-resolution sliding-spotlight Spaceborne SAR based on Antenna positioning vector. The method is technically sound and the experiments are sufficient. I have the following concerns.
1) The main idea of the paper is that the Range-Doppler equation (equation (5)) can not be directly used because of the dechirp step in sliding spotlight mode. So, the author uses equation (6) for calculation, because that after de-chirp all the Doppler centers become zero. However, after dechirp step the Doppler centers are not exactly zero because of the attitude errors of the satellite. This should be mentioned in the paper and provide discussions. Besides, equation (5) and (6) are not in the same coordinate I think, there is no VT here. It is better to use a unique coordinate in the paper.
2) In figure 8, “the orbit determination coordinate of SAR at the imaging time at target T is S”. Here, the imaging time at target T should be specified, because it is related to the imaging algorithm. So, the imaging method after dechirp step should be clarified.
Besides, the language and figures can be improved.
1) Title, “ground positioning method” is better than “ground positioning principle”
2) Abstract. Here in the squint spotlight mode, it is difficult to direct use the Range-Doppler equation for positioning because of the dechirp step. So, “strict Range-Doppler equation” is better to be “direct Range-Doppler equation”. There maybe typos in the abstract, “SAR load” may be “SAR payload”.
3) Introduction. “side view” is better to be “side-looking”.
4) The authors said that “after dechirp, the true Doppler center frequency of each point cannot be known”, which I think is not true. The dechirp signal is known, so the true Doppler center can be calculated.
5) Section II. There are many common variables where the authors using uncommon symbols, such as the speed of light “SOL” here, is better to use “c”, “FFS” here is better to be “fs”. Please check and revise to make the paper easy to read.
6) Equation (1), the R(\eta) should be r(\eta).
7) The location of the simulated targets should be plot before figure 1.
8) Figure 5 needs to be more standardized. Figure 6 can be improved too.
9) Figure 9 is not easy to understand, it can be improved.
Reviewer 2 Report
The manuscript (remotesensing-1893890-peer-review-v1) has been reviewed. The major comments and suggestions are in the enclosed file.

Reviewer 3 Report
The authors present an exciting paper about a conformation model based on an antenna pointing vector is presented which fully considers the influence of the dechirp operation on the range image, starts from the relative position of the dechirped range image points and the satellite, establishes a strict conversion model between image coordinates and geographic coordinates using the accurate satellite-ground geometric conditions. The manuscript is clear, relevant for the field and presented in a well-structured manner, and scientifically sound. The manuscript’s results are reproducible based on the details given in the methods section. The manuscript is well written and should be of great interest to the readers. However, all figures with charts could be more significant. Also the conclusion should mention more about their future work.
Reviewer 4 Report
This paper proposes a calculation of pointing vector for high resolution squinted SAR observation. The proposal mainly deals with pointing vector calculating from its orbital and positioning data i.e., the telemetry data. As long as this reviewer understands from Fig. 8 and the body text, those calculations seem to have been done during the traditional calibration. That is, we firstly believe the telemetry data and calibrate those parameters such as orbit, attitude, antenna position, direction and antenna pattern so that finally we can validate the geometric accuracy of SAR. It is difficult to find whether the authors did more advanced procedure.
Therefore, this reviewer suggests this article should be revised by clarifying the problem settings and their advances.
Applying sliding spotlight requires an accurate antenna steering vector while it is unclear from the article what data is unreliable or required to be calibrated. Positional error is a result and not a cause.
Experimental results are unclear. The authors didn’t show what SAR they used or what B1-B6 are. No readers can evaluate the experimental results. The authors have to compare with other focusing methods for showing superiority of the proposal as well.
Round 2
Reviewer 2 Report
The manuscript has been well-revised according to the comments provided. The revised manuscript clearly states the purpose of the study and proposes a logical method to achieve the objective.
Author Response
Dear expert,
Thanks for your comments concerning our manuscript. Thanks for your help.
Sincerely yours,
Yingying
Reviewer 4 Report
The revised manuscript seems fine except for some minor issues.
1. Please use proper word instead of side-looking angle. Is this off-nadir angle or depression angle?
2. Explanation of "Pulse-width and band-width combination ID" in Table 2 and 3 is hardly understandable. Please describe or refer proper citation.
